# MultiFacet: A Multi-Tasking Framework for Speech-to-Sign Language Generation

**Mounika Kanakanti***
International Institute of Information Technology
Hyderabad, India
mounika.k@research.iiit.ac.in
Mounika.Kanakanti@mpi.nl

**Shantanu Singh**
International Institute of Information Technology
Hyderabad, India
shantanu.singh@research.iiit.ac.in

**Manish Shrivastava**
International Institute of Information Technology
Hyderabad, India
m.shrivastava@research.iiit.ac.in

## ABSTRACT

Sign language is a rich form of communication, uniquely conveying meaning through a combination of gestures, facial expressions, and body movements. Existing research in sign language generation has predominantly focused on text-to-sign pose generation, while speech-to-sign pose generation remains relatively underexplored. Speech-to-sign language generation models can facilitate effective communication between the deaf and hearing communities. In this paper, we propose an architecture that utilises prosodic information from speech audio and semantic context from text to generate sign pose sequences. In our approach, we adopt a multi-tasking strategy that involves an additional task of predicting Facial Action Units (FAUs). FAUs capture the intricate facial muscle movements that play a crucial role in conveying specific facial expressions during sign language generation. We train our models on an existing Indian Sign language dataset that contains sign language videos with audio and text translations. To evaluate our models, we report Dynamic Time Warping (DTW) and Probability of Correct Keypoints (PCK) scores. We find that combining prosody and text as input, along with incorporating facial action unit prediction as an additional task, outperforms previous models in both DTW and PCK scores. We also discuss the challenges and limitations of speech-to-sign pose generation models to encourage future research in this domain. We release our models, results and code to foster reproducibility and encourage future research[1].

## CCS CONCEPTS

• **Computing methodologies** → **Neural networks**; **Learning latent representations**; **Computer vision**; **Information extraction**.

---

*Also with Max Planck Institute for Psycholinguistics, Nijmegen, Netherlands.

[1]https://github.com/Mounika2405/MultiFacet-Speech-to-Sign.git

---

## KEYWORDS

speech to sign language, indian sign language, prosody, pose generation

**ACM Reference Format:**
Mounika Kanakanti, Shantanu Singh, and Manish Shrivastava. 2023. Multi-Facet: A Multi-Tasking Framework for Speech-to-Sign Language Generation. In *INTERNATIONAL CONFERENCE ON MULTIMODAL INTERACTION (ICMI '23 Companion), October 9–13, 2023, Paris, France*. ACM, New York, NY, USA, 9 pages. https://doi.org/10.1145/3610661.3616550

## 1 INTRODUCTION

Sign language is a rich form of communication that seamlessly blends together the fluidity of hand movements and gestures, the expressiveness of facial expressions and head movements, and the subtle nuances of body language. It is this harmony of hand movements and expression that makes it complete and effective. According to the World Health Organization (WHO), over 1.5 billion people, which accounts for approximately 20% of the global population, live with hearing loss, underscoring the importance of accessibility in communication [15]. While the field of Natural Language Processing (NLP) has made remarkable progress in developing language technologies that simplify daily tasks, the advancement in technology to support sign language has not been as substantial [32]. Towards this end, automatic sign language translation and generation systems provide an efficient and accessible means of communication between deaf people and the hearing community.

Recent years have seen a surge of interest in sign language technologies, with researchers exploring various computer vision and deep learning approaches to tackle this complex task [17]. While many of these works utilize text or gloss as input for generation tasks, the area of speech-to-sign language generation remains relatively underexplored [17]. Gloss, often used to represent sign language, has been found to lack accuracy in capturing the complete linguistic and expressive aspects of sign language [29, 35]. A study on the Phoenix dataset [4] showed that a significant portion of the data contained linguistic elements not present in the gloss representation [35]. While text input can help generate semantic signs, incorporating prosodic information extracted from audio can provide more comprehensive data for a richer sign language output[7].

Inspired by co-speech gesture generation literature[14], which shares similarities with sign language generation, we utilize audio along with text as input to generate sign pose sequences. In this paper, we introduce MultiFacet, an architecture that uses prosodic information derived from speech and semantic information sourced

from text. This integrated data serves as the input for generating key points of both facial and hand movements. Furthermore, our approach includes the prediction of Facial Action Units (AUs) within a multi-tasking setup. We evaluate our model using Dynamic Time Warping (DTW) and Probability of Correct Keypoints (PCK) metrics against the existing Indian Sign Language dataset[11] and demonstrate the critical importance of prosody and facial action unit prediction in better sign language generation. In summary, our main contributions are as follows:

- Leveraged prosody information from the audio and semantic context from text for generation of continuous sign pose sequences.
- Exploring the importance of facial action unit prediction for generating hand and face key points in Indian sign language.
- We conducted ablation studies and extensively discussed the limitations of our work to inspire future research and advancements in this domain.

## 2 RELATED WORK

**Sign Pose Generation** Most of the works in sign language generation are based on text or gloss as inputs[17]. [20] generated continuous hand pose sequences using text as input. While this is a great step in the field, it is only a partial representation of sign language, as facial expressions and body language also play a critical role in conveying meaning [9, 16]. Later works attempted to address this limitation by including both manual (hand movements) and non-manual (facial expressions) features in the generation process but still relied on text or gloss as input. [18] used adversarial training for multichannel sign production with text as input. Furthermore, in another study, [21] represented sign sequences as skeletal graph structures with gloss as an intermediate representation. [28] generated key points for hands and face by concatenating embedding outputs from a text encoder and a gloss encoder. [29] first generated Hamnosys notation from text, which was further converted to continuous sign pose sequences. These approaches made strides towards incorporating non-manual features but still lacked the use of prosodic information as input corresponding to the non-manual features in sign language. [19, 25] generated photo-realistic sign videos using text as input. They first generated skeleton poses from text and then generated sign videos conditioned on these poses.

To address this concern of loss of prosody in gloss representation, [35] presented gloss enhancement strategies for introducing intensity modifiers in gloss annotations using Phoenix dataset [4]. Intensity modifiers are the ones that quantify nouns, adjectives or adverbs in a sentence ((e.g., very happy or little happy). Recent works explored the use of speech Mel spectrogram inputs to generate hand movements in Indian Sign Language [11]. While this approach is a step in the right direction, generating hand movements alone is insufficient to capture the full extent of sign language.

**Co-speech Gesture Generation** Co-speech gesture generation studies have shown the significance of using both speech and text as input for generating semantically relevant and rythmic gestures [14]. [1, 12, 33] have proposed continuous gesture generation systems using audio and text as input, further underscoring the importance of multimodal information for generating meaningful gestures.

**Non-Manual Recognition in Sign Language** [26] presented 3D-CNN based multimodal framework for recognition of grammatical errors in continuous signing videos belonging to different sentence types. The methodology they employed encompassed two primary stages. Initially, 3D-CNN networks were leveraged to recognise the grammatical elements from manual gestures, facial expressions, and head movements. Subsequently, a sliding window technique was adopted to establish correlations between these modalities, thereby facilitating the detection of grammatical errors in the signing videos.

In this paper, by incorporating prosody and non-manual feature recognition, such as predicting Facial Action Units, we aim to improve the accuracy and naturalness of sign language generation.

## 3 SPEECH TO SIGN LANGUAGE GENERATION

Given audio and text inputs, our aim is to generate sequences of sign poses denoted as S, which include both upper body and face keypoints. To accomplish this, we adopt a multi-task learning approach, incorporating a speech encoder, a Facial Action Units decoder, and a sign pose decoder. The overall architecture is illustrated in Figure 1.

### 3.1 Input Embeddings

To facilitate the generation process, we extract two types of embeddings from the input data: BERT embeddings for text and Tacotron 2 GST[30] encodings for audio. We use the GST model provided by NVIDIA[2] which was pre-trained on *train-clean-100* subset of LibriTTS dataset[34] to represent the expressive features in audio. The BERT embeddings, denoted as $E_{text}$, capture the semantic information embedded within the text, allowing our model to understand the linguistic context. We represent the input text as a sequence of tokens $\{x_1, x_2, ..., x_W\}$, and BERT provides the corresponding embeddings $\{e_{x_1}, e_{x_2}, ..., e_{x_W}\}$ with a dimensionality of 768.

The Tacotron 2 GST encodings, denoted as $E_{audio}$, extract both linguistic content and prosody information from the audio input. The GST model was pre-trained on LibriTTS dataset [34] with the objective of learning a large range of acoustic expressiveness. We represent the audio input as a sequence of mel-spectrograms $\{m_1, m_2, ..., m_T\}$, where each mel-spectrogram has $T \times 256$ dimensions. Tacotron 2 GST[30] provides the corresponding embeddings $\{e_{m_1}, e_{m_2}, ..., e_{m_T}\}$.

### 3.2 FAUs Preprocessing

Amongst various methods for denoting facial expressions, the Facial Action Coding System (FACS) stands as a comprehensive and standardized tool [31]. It has been meticulously designed to describe and analyze these nonverbal cues by precisely identifying distinct facial muscle movements. Central to FACS are its action units (FAUs), a set of codes representing individual facial muscle actions, which, when combined, proficiently portray a diverse array of emotions and expressions. As a result of its efficacy, FACS finds widespread application across various disciplines, including psychology, neuroscience, anthropology, and computer graphics, providing an objective and systematic means to categorize and comprehend facial expressions.

---

[2]https://github.com/NVIDIA/mellotron/tree/master

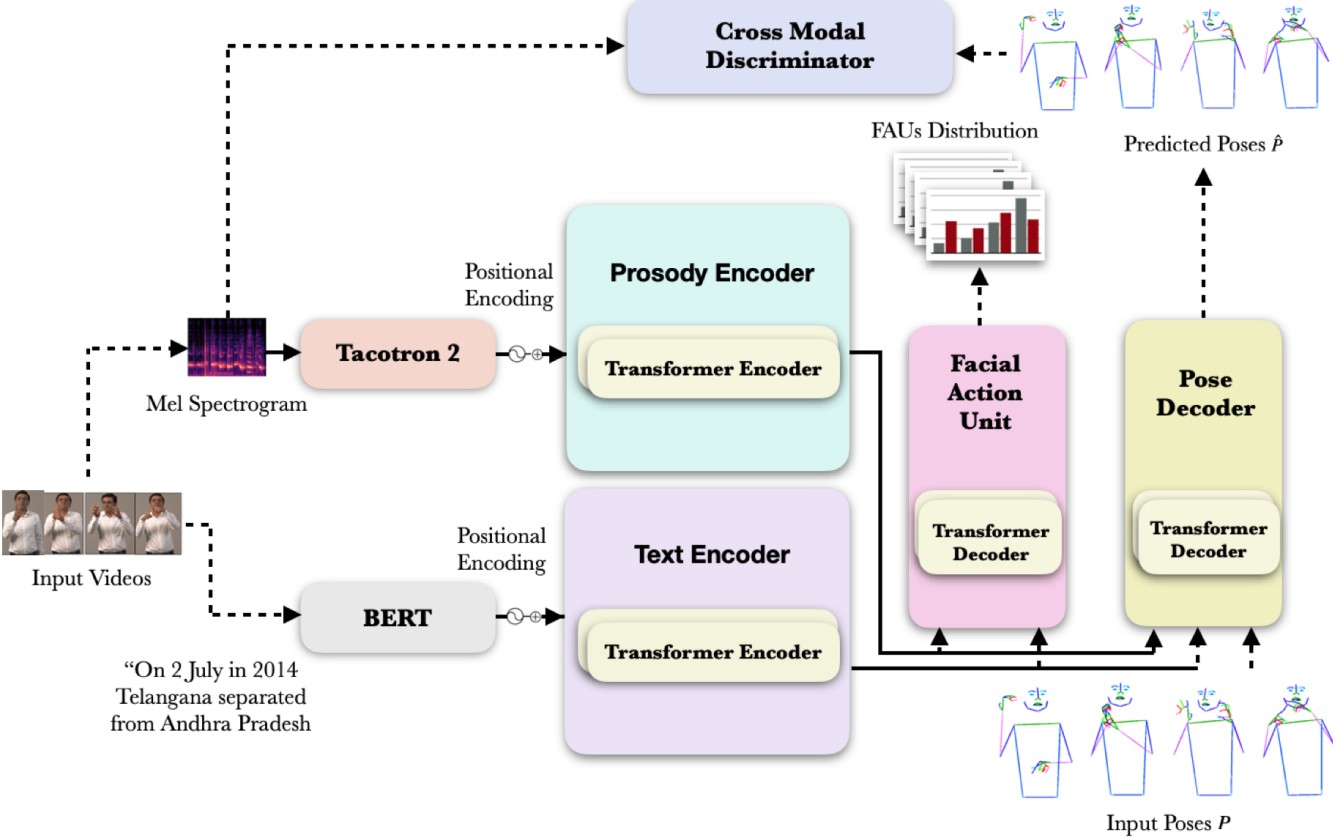

**Figure 1: The Architecture: We propose a novel architecture to generate sign pose sequences by utilising the prosodic information from speech and semantic context from text. We also incorporate additional decoders to facilitate rich sign pose generation: (i) Facial Action Unit decoder and (ii) Cross modal decoder.**

While FACS is an index of facial expressions with an anatomical basis, it generally does not provide the degree of muscle activation. While there are modifiers that extend this coding system to accommodate intensities as well, we don't consider them in our study due to limited resources and no clear consensus on their use.

The use of FACS for sign language translation or generation is relatively understudied [5, 6, 23]. One of the primary reasons for its limited use is the costly annotation required for the existing sign language datasets. To overcome this issue, we propose using an existing state-of-the-art model, ME-GraphAU [13], to predict the action units for our chosen dataset and use it as weak-supervision during sign-language generation task. We encourage readers to refer to [13] for details related to architecture, training dataset and output format for the aforementioned model.

The output of the chosen model is noisy and lacks temporal consistency since the prediction occurs on a per-frame basis. Training with such an output would invariably lead to noisy supervision and poor learning on the model's part for the proposed task. As such, we propose a pre-processing pipeline for reducing the noise using the following steps:

- Threshold the output of the model using the probabilities as confidence for each action unit and remove any low confidence predictions.
- For these pruned predictions, we use linear interpolation for estimating their new values.
- Finally, to reduce the remaining noise, we use hanning smoothing over each action unit and get the final output. We use a window length of 11, which corresponds to 0.5 seconds at 24FPS frame-rate of our source videos.

We show an example of the original prediction and output of each step in the above-mentioned pipeline in Figure 2.

Figure 3 shows the ground truth facial action units extracted.

### 3.3 Model Components

The input embeddings $\mathbf{E}_{\text{text}}$ and $\mathbf{E}_{\text{audio}}$ are then passed to their respective encoders in our model:

1. **Prosody Encoder**: The transformer-based speech encoder, denoted as $E_{\text{speech}}$, processes the Tacotron 2 GST encodings $\mathbf{E}_{\text{audio}}$ to obtain intermediate representations $\mathbf{H}_{\text{speech}}$. This can be expressed as:

$$\mathbf{H}_{\text{speech}} = E_{\text{speech}}(\mathbf{E}_{\text{audio}})$$

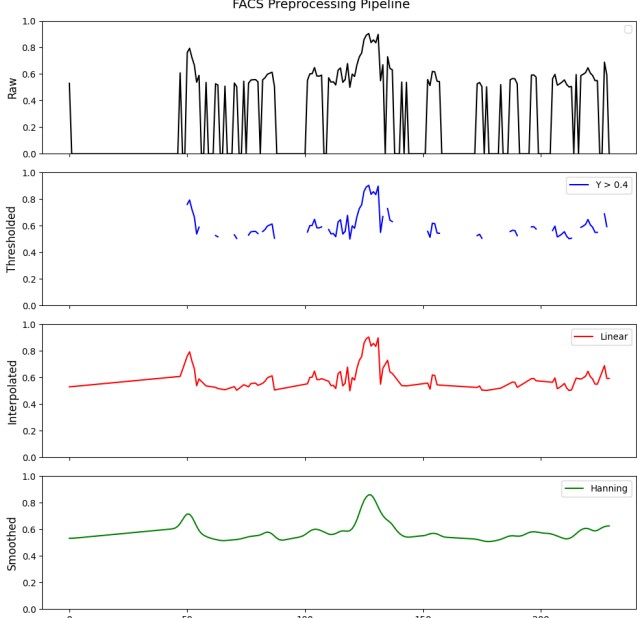

**Figure 2: Illustration of the Facial Action Units (FAUs) preprocessing pipeline: thresholding using action unit probabilities, linear interpolation, and Hanning smoothing.**

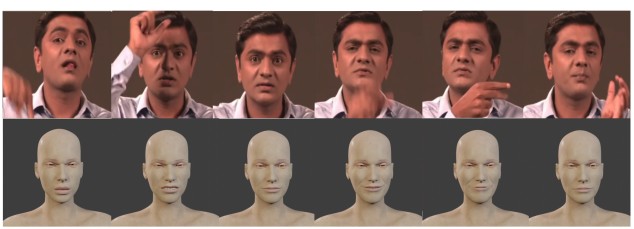

**Figure 3: Representation of Ground Truth Facial Action Units, generated using Blender[3] for visualization purposes.**

2. **FAUs Decoder**: We incorporate the FAUs prediction task as an additional objective to capture facial expressions. The FAUs decoder, denoted as $D_{\text{FAUs}}$, processes the Tacotron 2 GST encodings $\mathbf{E}_{\text{audio}}$ to predict the Facial Action Units, denoted as **FAUs**. This can be expressed as:

$$\mathbf{FAUs} = D_{\text{FAUs}}(\mathbf{E}_{\text{audio}})$$

Facial AUs is a widely used facial expression coding system that consists of a set of action units that correspond to different facial muscle movements. We use a transformer-based decoder[27] for this task and train it using cross-entropy loss.

$$\mathcal{L}_{\text{FAUs}} = -\frac{1}{N} \sum_{n=1}^{N} \sum_{i=1}^{M} y_{n,i} \log(p_{n,i}) \tag{1}$$

where $N$ is the number of training examples, $M$ is the number of Facial Action Units, $y_{n,i}$ is the ground-truth label for the $i$-th Facial Action Unit in the $n$-th example (either 0 or 1), and $p_{n,i}$ is

the predicted probability for the $i$-th Facial Action Unit in the $n$-th example.

3. **Sign Pose Decoder**: Our sign pose decoder, denoted as $D_{\text{pose}}$, is a transformer-based autoregressive decoder that takes the intermediate representations $\mathbf{H}_{\text{speech}}$ as input to generate the sequence of sign poses $\mathbf{S}$. The keypoints for each frame in the sign pose sequence are represented as a 3D tensor, with dimensions num_frames $\times 85 \times 3$. The output of the decoder can be formulated as:

$$\hat{y}_{n,i} = \mathbf{D}_{\text{Pose}}(\mathbf{H}_{\text{speech,n}}, \mathbf{y}_{n,0:i-1}) \tag{2}$$

Note that during training, the decoder uses ground-truth poses as input for stability and faster convergence. During inference, the pose inputs to the decoder are its own predictions upto the given timestep.

We use regression loss to train the sign pose decoder, given by:

$$\mathcal{L}_{\text{pose}} = \frac{1}{N} \sum_{n=1}^{N} \sum_{i=1}^{85} \|y_{n,i} - \hat{y}_{n,i}\|^2 \tag{3}$$

where $N$ is the number of training examples, $y_{n,i}$ is the ground-truth value of the $i$-th keypoint for the $n$-th example, and $\hat{y}_{n,i}$ is the predicted value of the $i$-th keypoint for the $n$-th example.

4. **Cross-Modal Discriminator** We use the same discriminator used by [11] to match the speech segments with corresponding pose sequences. The loss for the cross-modal discriminator can be defined as follows:

$$\mathcal{L}_{\text{G}}^{\text{GAN}} = \frac{1}{N} \sum_{n=1}^{N} \log(1 - (\mathbf{D}_{\text{cross-modal}}(\mathbf{H}_{\text{speech, n}}, \hat{\mathbf{y}}_n))) \tag{4}$$

$$\begin{aligned}\mathcal{L}_{\text{D}}^{\text{GAN}} = -\frac{1}{N} \sum_{n=1}^{N} &\log((\mathbf{D}_{\text{cross-modal}}(\mathbf{H}_{\text{speech, n}}, \mathbf{y}_n))) \\ &+ \log(1 - (\mathbf{D}_{\text{cross-modal}}(\mathbf{H}_{\text{speech, n}}, \hat{\mathbf{y}}_n)))\end{aligned} \tag{5}$$

where $\mathbf{D}_{\text{cross-modal}}$ is the cross-modal discriminator. $\mathbf{H}_{\text{speech, n}}$ is the intermediate representation for the $n$-th example obtained by the prosody encoder. Variables $\mathbf{y}_n$ and $\hat{\mathbf{y}}_n$ are the ground-truth and predicted pose sequences respectively. $\mathcal{L}_{\text{D}}^{\text{GAN}}$ and $\mathcal{L}_{\text{G}}^{\text{GAN}}$ are the standard binary cross-entropy loss used for discriminator and generator respectively.

### 3.4 Multi-Tasking Setup

We use a weighted sum of the losses from the individual decoders to compute the overall loss.

$$\mathcal{L}_{\text{total}} = \lambda_{\text{FAUs}} \cdot \mathcal{L}_{\text{FAUs}} + \lambda_{\text{pose}} \cdot \mathcal{L}_{\text{pose}} + \lambda_{\text{discriminator}} \cdot \mathcal{L}_{\text{G}}^{\text{GAN}}$$

where $\lambda_{\text{FAUs}}$, $\lambda_{\text{pose}}$, and $\lambda_{\text{discriminator}}$ are hyperparameters that control the relative importance of the FAUs loss, pose loss, and discriminator loss, respectively.

The weights for each task are chosen to balance the contribution. All the decoders are trained in a multitasking setup. The model is trained to minimize the multitasking loss $\mathcal{L}_{\text{total}}$ using gradient-based optimization techniques.

## 4 IMPLEMENTATION DETAILS

We set up our transformer model with two layers for both encoders and decoders, each equipped with eight attention heads. Both encoders and decoders use a hidden size of 512. We use the Adam optimiser with an initial learning rate of 0.001, which can be reduced if the training plateaus. We apply gradient clipping with a threshold of 5.0 and use a batch size of 32 for training efficiency. We incorporate Future Prediction as proposed by [20]. The training loss function includes L1 regularisation along with losses for specific components, each weighted accordingly. For the loss function, the values for $\lambda_{\text{Pose}}, \lambda_{\text{FAUs}}, \lambda_{\text{Discriminator}}$ are 1, 0.001, 0.0001 respectively.

## 5 EXPERIMENTS

### 5.1 Dataset

The dataset used in our study is the continuous Indian Sign Language dataset, which was released by [11]. This dataset contains sign videos along with corresponding audio and text transcription, covering various topics, such as current affairs, sports, and world news. The dataset comprises 9137 videos and has a vocabulary size of 10k.

To represent the sign videos in our analysis, we extracted 3D joint position keypoints using Mediapipe [8]. This process involved detecting 37 landmark points for the eyes, eyebrows, lips, and face outline, along with 6 landmark points for the shoulders, elbows, and hips. Additionally, each hand was represented with 21 landmark points, bringing the total to 85 keypoints for upper body, hands and face.

### 5.2 Baseline Models

**Text2Sign** We adopt the progressive transformers introduced by [20] as the foundation of our approach. We extend their proposed architecture and train them on the Indian Sign Language Dataset with 3D keypoints for face and upper body.

**Speech2Sign** [11] utilised mel spectrograms as input to generate sign pose sequences of hand movements. They incorporate a text decoder and a cross-modal discriminator for learning the correlation between speech and sign pose sequences. We extend this architecture to generate face and body key points and consider it as our baseline.

### 5.3 Evaluation Metrics

**Dynamic Time Warping (DTW)** Dynamic Time Warping (DTW) [10] is one of the evaluation metrics for speech-to-sign language generation models to assess the alignment between the predicted sign language sequences and the ground truth sign language sequences.

Let $P = (p_1, p_2, \ldots, p_M)$ denote the predicted sign language sequence, where $p_i$ represents the $i$-th pose in the predicted sequence, and $M$ is the length of the predicted sequence. Similarly, let the ground truth sign language sequence be denoted as $G = (g_1, g_2, \ldots, g_N)$, where $g_i$ represents the $i$-th pose in the ground truth sequence, and $N$ is the length of the ground truth sequence.

DTW aims to find an optimal alignment between the sequences $P$ and $G$ by introducing a warping path $W = \{(w_1, w_2, \ldots, w_K)\}$,

where $w_k = (i, j)$ denotes the alignment of $p_i$ in the predicted sequence with $g_j$ in the ground truth sequence. The warping path satisfies the conditions: $w_1 = (1, 1)$, $w_K = (M, N)$, and $w_k - w_{k-1} \in \{(1, 0), (0, 1), (1, 1)\}$, allowing for insertions, deletions, and matches between the sequences.

The objective of DTW is to minimize the accumulated cost along the warping path $W$, which is defined by a distance or similarity measure between the individual poses in the sequences. Let $d(p_i, g_j)$ represent the distance between $p_i$ and $g_j$ in the pose space. The accumulated cost $C(W)$ along the warping path $W$ is given by:

$$C(W) = \sum_{k=1}^{K} d(p_{w_k}, g_{w_k})$$

To compute the final DTW score, we aim to find the optimal warping path $W^*$ that minimizes the accumulated cost $C(W)$:

$$DTW(P, G) = \min_{W} C(W)$$

The DTW score provides a measure of the alignment between the predicted and ground truth sign language sequences, considering the temporal differences and variations in the movement patterns. A lower DTW score indicates a better alignment and higher similarity between the sequences.

**Probability of Correct Keypoints (PCK)** PCK [2, 24] is a widely used evaluation metric to assess the accuracy of pose estimation models. It measures the percentage of correctly predicted keypoints within a certain threshold distance compared to the ground truth keypoints.

Let $G = \{g_1, g_2, \ldots, g_N\}$ be the set of ground truth keypoints, and $P = \{p_1, p_2, \ldots, p_N\}$ be the set of predicted keypoints. Each keypoint, $g_i$ or $p_i$, consists of $(x, y, z)$ coordinates representing the position of a particular body part, such as a hand or face.

To compute the PCK score, we need to define a threshold distance $\delta$. For each ground truth keypoint $g_i$, we check if there exists a corresponding predicted keypoint $p_j$ within the threshold distance $\delta$. If such a predicted keypoint exists, and its distance to the ground truth keypoint is less than or equal to $\delta$, we consider it as a correct prediction.

Mathematically, the PCK score can be computed as follows:

$$PCK = \frac{1}{N} \sum_{i} \delta(g_i, p_i)$$

where $N$ is the total number of keypoints, and $\delta(g_i, p_i)$ is an indicator function defined as:

$$\delta(g_i, p_i) = \begin{cases} 1, & \text{if } ||g_i - p_i|| \leq \delta \\ 0, & \text{otherwise} \end{cases}$$

Here, $||g_i - p_i||$ represents the Euclidean distance between the ground truth keypoint $g_i$ and the predicted keypoint $p_i$.

The PCK score is then calculated as the average of the indicator values over all keypoints. It represents the percentage of keypoints that have been correctly predicted within the specified threshold distance $\delta$. A higher PCK score indicates better accuracy and alignment between the predicted and ground truth keypoints.

In the context of sign language generation models, PCK can be used to evaluate the quality of the generated sign language poses by

**Table 1: Comparison of Dynamic Time Warping (DTW) and Probabilty of Correct Keypoints (PCK) scores with baselines on dev and test sets. B+F indicates model that predicts body+face keypoints. PE - Prosody Encoder; TE: Text Encoder**

| Model | DTW Score ↓ | PCK ↑ |
|---|---|---|
| Dev set | | |
| Text ->Sign[20] | 19.55 | 0.61 |
| Speech2sign [11] | 15.94 | 0.72 |
| PE + TE ->Sign | 16.1 | 0.74 |
| PE + TE ->Sign + FAUs | **13.37** | **0.79** |
| Test set | | |
| Text ->Sign[20] | 22.55 | 0.59 |
| Speech2sign [11] | 14.08 | 0.78 |
| PE + TE ->Sign | 17.3 | 0.72 |
| PE + TE ->Sign + FAUs | **13.23** | **0.81** |

comparing them to the ground truth poses. However, it's important to note that PCK only considers individual keypoints and does not capture the overall spatial or temporal coherence of the generated sign language sequences.

## 5.4 Results and insights

We report DTW[10] and Probability of Correct Keypoints scores on the Indian Sign Language dataset and compare it with the results of both Text2Sign[20] and Speech2Sign [11] methods. From table 1 we observe that our model performs significantly better than the existing Speech2Sign[11] method. Figure 4 shows the sample qualitative results. An interesting observation from the provided sample results, as well as other instances in our evaluation, is that while our model encounters challenges in accurately capturing the precise positions of hands and facial features in specific frames, these representations exhibit a visual similarity to the target RGB frames. It is worth noting, however, that minor disparities in hand positions and facial expressions can convey substantially different meanings in sign language. Consequently, we refrain from drawing definitive conclusions from our qualitative assessments and defer such considerations to future research endeavors.

## 6 ABLATION ANALYSIS

To evaluate the contribution of each component in our proposed architecture, we conduct ablation studies on our model. Specifically, we perform experiments where we remove each component from the multitasking setup one by one and compare the results with the full model.

Table 2 summarizes the results of our ablation studies. As can be seen, removing the FAUs decoder results in a drop in performance in both metrics. The results demonstrate the effectiveness of our multitasking approach in leveraging multiple modalities for sign language generation. However, we observe that the results are still close to the model that uses only the text encoder.

In summary, our ablation studies demonstrate the effectiveness of our multitasking approach in leveraging multiple modalities for sign language generation.

**Table 2: Comparison of ablation studies. PE - Prosody Encoder; TE-Text Encoder**

| Model | DTW Score ↓ | PCK ↑ |
|---|---|---|
| TE ->Sign | 13.82 | 0.81 |
| TE ->Sign + FAUs | 15.69 | 0.78 |
| PE ->Sign | 17.16 | 0.73 |
| PE ->Sign + FAUs | 14.52 | 0.75 |
| PE + TE ->Sign + FAUs (Ours) | **13.23** | **0.81** |

## 7 LIMITATIONS & CHALLENGES

**Evaluation Methods:** Although our model has achieved state-of-the-art results based on DTW scores, it is essential to conduct human evaluation with expert sign language interpreters to ensure the quality and relevance of the generated sign language. DTW scores only assess the alignment between ground truth poses and predicted poses but do not measure the correlation with the input speech. Correlating these scores with human evaluation ratings is crucial for understanding the model's performance in real-world communication scenarios. Metrics that measure the coherence and synchronization of other non-manual elements, such as body posture, head movements, and eye gaze are also necessary [26]. Therefore, when designing a sign language generation model, accounting for these linguistic elements and their dynamic interactions is essential to produce more accurate and culturally appropriate sign language outputs.

**Fine Movements:** The current model successfully learns coarse hand movements but lacks the ability to capture fine movements of fingers and facial parts (See Figure 5 in Appendix A)s. This limitation is attributed to the use of Mean Squared Error (MSE) loss, which penalizes larger movements more than fine movements. To address this issue, alternative loss functions, such as a keypoint loss proposed by [22], can be explored. This loss involves a hand keypoint discriminator pre-trained on 2D hand poses and may improve the model's capability to generate more accurate and intricate hand movements.

**More Linguistic Information:** One significant challenge lies in handling the sequential nature of input speech or text, as opposed to the simultaneous nature of sign language. Speech unfolds in a linear manner, and sign language relies on the integration of multiple components in parallel. Thus, capturing and mapping these linguistic structures effectively requires specialized attention. Understanding how signers use space, directionality, and facial expressions to indicate different grammatical constructs is crucial for generating natural and contextually appropriate sign language. Currently, our model focuses primarily on generating hand and facial movements, neglecting other crucial components. Future work should explore incorporating non-manual markers, body language, and gaze direction into the generation process to enhance the naturalness and comprehensiveness of sign language communication.

**Errors in Skeleton Pose Extraction:** One of the significant challenges in sign language generation is accurately extracting the

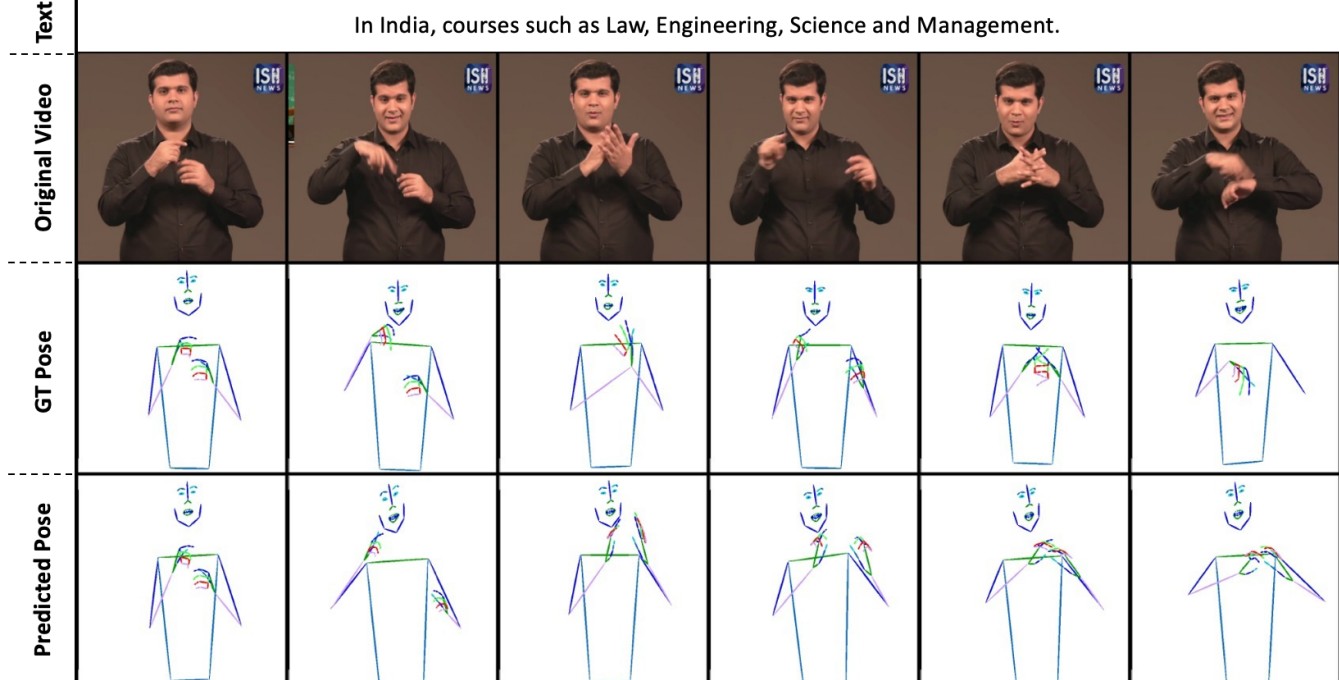

**Figure 4: Qualitative Results illustrating the input text, the original video, the ground truth pose, and the predicted pose.**

skeleton pose from the input video or speech. The skeleton pose serves as a crucial input to the model, representing the keypoint positions of the signer's hands, face, and body movements. Although advanced pose estimation techniques like Mediapipe provide robust keypoint predictions, there are inherent limitations and errors that can impact the overall performance of the sign language generation model. Sign language videos captured in real-world settings may contain various forms of noise, occlusions, and artifacts. These imperfections can lead to inaccuracies in the pose estimation process, resulting in incorrect keypoint positions. For instance, background clutter, complex hand gestures, or fast movements may obscure the hand keypoints, leading to incomplete or noisy pose representations. Additionally, sign language involves intricate hand and finger movements that can sometimes be challenging to discern accurately (See Figure 6 in Appendix A ). The dynamic nature of sign language requires precise identification of hand shapes, finger positions, and gestures. However, the inherent ambiguity in certain signs or gestures can lead to misinterpretations and inaccuracies in the extracted skeleton pose.

**Pose Representation:** The representation of sign language as keypoint sequences in videos is abstract and results in the loss of some skeletal information. This may lead to some loss of fine-grained details in the generated sign language. Future research could explore alternative representations that preserve more intricate skeletal information for more accurate sign language generation.

**Dataset Size and Variety:** Our current dataset size and variety might be limited, which could impact the model's ability to capture the full complexity and richness of sign language. Expanding the dataset or exploring low-resource training techniques is essential to improve the model's generalization and performance on diverse signing styles and linguistic patterns.

**Signer Style:** Sign language relies on the signer's individual style and preferences, which can significantly affect the model's performance. Investigating the impact of varying signer styles on the model's output and devising methods to adapt the model to different signing styles are critical for real-world applicability.

In conclusion, while our model shows promising results in generating sign language from speech, there are several limitations and challenges that need to be addressed in future work.

## 8 CONCLUSION

In this paper, we introduced a multi-tasking approach, the Multi-Facet model, for generating sign language poses from input speech and text. Our model goes beyond just hand movements, also capturing facial expressions, resulting in a more comprehensive representation of sign language.

To assess the effectiveness of our model, we conducted experiments on the Indian Sign Language dataset provided by [11]. By incorporating a pre-trained prosody encoder and utilizing Facial Action Units, we achieved even better results, surpassing previous methods. The potential applications of our approach extend beyond sign language communication.

Although we achieved better results with the proposed approach, there is significant room for further advancements in several aspects, including the datasets, methodologies, understanding of the intricate relationship between speech and sign language, and evaluation methods. We hope that our work will inspire further research in this area and contribute to improving accessibility and inclusivity for the deaf and hard-of-hearing community.

## 9 ETHICAL CONSIDERATIONS

In our study, it is important to acknowledge that we have employed a limited dataset of Indian sign language videos, primarily sourced from YouTube. While this dataset served as a valuable starting point for our investigation into speech-to-sign language generation models, we recognise its inherent limitations regarding representativeness for the broader sign language community. It is essential to emphasize that the models proposed in this paper are only to explore the role of prosody in speech-sign language generation models and are not suitable for direct deployment due to their insufficient scope and potential biases. Moreover, we acknowledge that a critical aspect, validation with signers, has not been fully undertaken within the scope of this study. This is a significant limitation that warrants further attention and validation in future research endeavours.

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

## A QUALITATIVE RESULTS

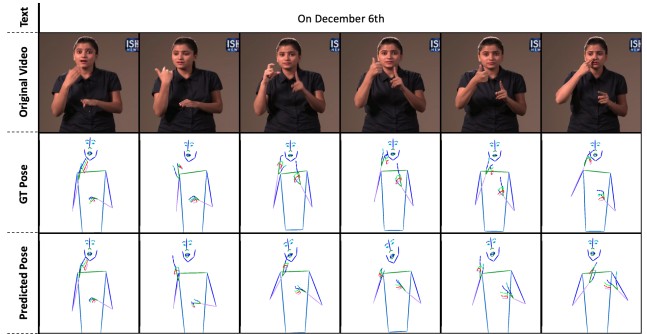

Figure 5: Sample result showing the model's accurate hand movement prediction with inaccurate finger movements.

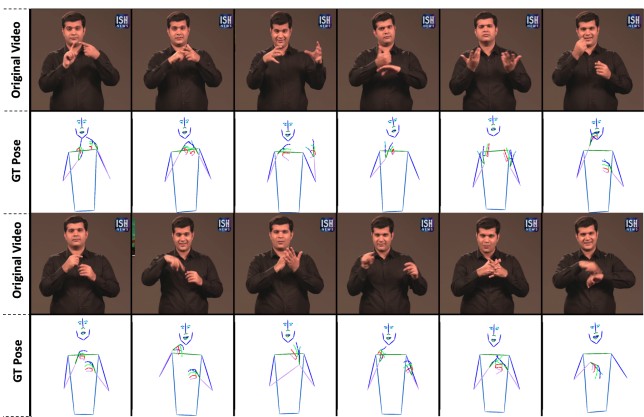

Figure 6: Mediapipe Errors. The keypoints for the fourth frame in the first video and the sixth frame in the second video are predicted incorrectly due to fast/blurry movements whereas the keypoints for the third frame in the second video are predicted incorrectly as it contains a complex hand gesture.