# OpenReview forum: "MultiFacet: A Multi-Tasking Framework for Speech-to-Sign Language Generation"
_ACM.org/ICMI/2023/Workshop/GENEA — GENEA Workshop 2023_

### Official Review · Reviewer_fX4K · 2023-07-28
**MultiFacet: A Multi-Tasking Framework for Speech-to-Sign Language Generation**

**Rating:** 9
**Confidence:** 4

**Review:**

The paper addresses a very challenging topic: Mapping speech to sign language.

Moreover, the authors do not focus solely on hand gestures but also on facial expressions.

The proposed behavior generation architecture is very detailed but it looks fine.

The evaluation studies rely on objective measures. So far, the authors use 2 measures but they present a list of other measures. It will be very interesting to hear how the authors intend to implement them, in particular regarding the ‘more linguistic information’.

No perceptual experiments are foreseen. Building readable signs by hearing-impaired participants is a challenging goal to attain.

---

### Official Review · Reviewer_uvia · 2023-08-06
**Interesting work but validation is quite limited**

**Rating:** 5
**Confidence:** 3

**Review:**

This paper introduces a novel transformer-based framework designed to generate Indian Sign Language (ISL) from both speech audio and speech text inputs. The authors employ an encoding process that utilizes Tacotron-GST and BERT, converting the speech audio and text into latent features. This innovative approach enables the representation of essential prosodic and semantic information contained within the raw data. The model's training process involves the use of 3D joint coordinates, consisting of 85 keypoints that cover the upper body, hands, and face. These keypoints have been extracted from a substantial dataset containing 9137 sign language video clips. The model's effectiveness has been evaluated using Dynamic Time Warping (DTW) distance and a metric called Probability of Correct Keypoints (PCK)-based similarity. The authors present a comparative analysis of the proposed method against two previously established techniques. The results demonstrate that the proposed method exhibits a slight improvement in performance.

It is plausible that the proposed approach enhances the communicative spectrum of sign language-based interaction by leveraging semantic and affective cues contained in the multimodal inputs. The approach generates not only symbolic hand poses but also facial expressions and body poses. However, a major issue I have with the paper is the lack of detailed presentation of the results, which makes it impossible to properly determine the validity and efficacy of the method. Here are the comments I would like to provide for the improvement of the paper:

1) Qualitative Evaluation Required: Relying solely on quantitative measures is very limited in the context of communicative gesture generation. Qualitative evaluation involving user participation should be conducted to support the efficacy of the method for communication. This is especially pertinent for this paper, as it proposes to enhance sign language-based communication by generating intricate facial expressions and body poses.

2) Provide Sample Results: Visualizations of generated results within the manuscript or a supplemental video would have greatly enhanced the reader's ability to appreciate the contributions of the paper. At the very least, the paper should have included detailed figures of generated hand and finger poses.

3) Quantify the Quality of Training Data: As the authors described in Section 7, the training data might contain quite a lot of errors, especially concerning finger motions. The extent of errors in the training data and their effect on the performance of the trained model should be examined and quantified.

The manuscript has several typos and citation errors. Please perform thorough proofreading to ensure clarity and accuracy.

---

### Decision · Program_Chairs · 2023-08-11

**Decision:**

Accept

**Comment:**

While this paper has some limitations as pointed out by the reviewer, the chairs agree that speech-to-sign, not from gloss or only text, is a new and challenging task and this paper did a good job in describing their system. The chairs think this paper is above the bar for a workshop track so we accept this paper for the GENEA Workshop.

Please read the reviews and update the paper.
- You should include the videos showing both human sign motion and generated results for visual inspection.
- Proofreading is needed.